# Validity and Reliability of a Novel Instrument for the Measurement of Subtalar Joint Axis of Rotation

**DOI:** 10.3390/ijerph18105494

**Published:** 2021-05-20

**Authors:** Byong Hun Kim, Sae Yong Lee

**Affiliations:** 1Department of Physical Education, Yonsei University, Seoul 03722, Korea; bh_kim@yonsei.ac.kr; 2International Olympic Committee Research Centre Korea, Yonsei University, Seoul 03722, Korea; 3Institute of Convergence Science, Yonsei University, Seoul 03722, Korea

**Keywords:** subtalar joint, subtalar joint axis of rotation, device development, validation studies

## Abstract

Inclination of the subtalar joint (STJ) in the sagittal and transverse planes may be highly associated with ankle pathology. However, the validity and reliability of measuring the inclination of the STJ axis of rotation (AoR) is not well established. This study aimed to develop a custom-made STJ locator (STJL) and evaluate its reliability and validity. To establish the reliability and validity of the measurement device for STJ AoR, 38 healthy male participants were recruited. For the reliability analysis, test–retest was used, and for validity analysis, Pearson’s correlation and Bland–Altman plot analyses were performed. In the reliability analysis of the STJL, a higher correlation was observed with the sagittal plane (0.930) and transverse plane (0.748) (standard error of measurement: 0.56–0.78; minimal detectable difference: 1.57–2.16). In the validity analysis between radiography and STJL, a significantly higher value of 0.798 was obtained with radiography (42.5) and STJL (43.5) with the sagittal plane. The custom-made STJL may be used in the clinical setting as its validity and intraclass correlation coefficient were high, indicating consistent measurements. Further studies including motion analysis are necessary to provide more information regarding the relationship between STJ AoR inclinations and STJ movements.

## 1. Introduction

The movement of the subtalar joint (STJ) is a complex three-dimensional motion that generates pronation and supination in the closed kinetic chain movements [1]. All movement types are determined by the joint axis of rotation (AoR). Considering the anatomical structures of the foot, muscle and ligament movements and ground reaction forces in the STJ are settled by the location and direction of the STJ axis [2]. Since the positions of the axis are significantly different from individual to individual, determining the position of the STJ axis of each patient is an important objective [3,4,5,6]. Although the great variability in the actual angle of the STJ axis is functionally important in the sagittal, transverse, and frontal planes, it has been scarcely referred to or researched in studies.

Clinically, locating the AoR of the STJ is important for assessing the position and motion of the hindfoot. Kirby et al. have reported that the medial or lateral deviation in the spatial location of the STJ AoR can have significant effects on the kinematics of the lower extremity during weight-bearing (WB) activities [1]. Various impairments, including rearfoot varus and valgus deformities, are related to decreased muscular strength that rotates the STJ [7]. The results of a study suggested that the STJ AoR plays an important role in maintaining the rotational balance of the foot during corresponding movements of the foot in WB conditions.

There have been diverse attempts to measure the STJ AoR. In the initial stage, the STJ was measured with the use of non-weight-bearing conditions using cadavers [2,3] only in in vivo studies; this has now progressed to WB conditions also being used. However, they were invasive and accompanied by movements of other joints [8]. McClay has measured the STJ AoR using four methods on the sagittal plane, and the results showed a relatively high level of validity [4]. It showed a close correlation in three of the four cadaveric studies [9]. Moving the foot with minimal talocrural movement around the STJ might be possible to estimate the STJ from the helical axis of tibiocalcaneal movements.

Although researchers have attempted using several methodologies for years to determine the tracking of STJ motion, the literature reveals no clinically viable methods that allow evaluation of STJ AoR in three dimensions during WB conditions [1,3,10,11,12,13,14,15,16,17,18,19,20,21,22,23,24,25,26,27,28].

Since tracking the talus is impossible using skin-mounted markers, the location of the STJ AoR is difficult to be measured in vivo [5]. Given the difficulty in tracking the movements of the talus in vivo, only the overall motion of the ankle–STJ complex, that is, the relative motion of the foot with respect to the shank, was measured in this study [6]. A limitation of this typical representation is that it does not discern how movements affect the articulation of the tibiotalar joint and STJ [29,30,31]. For this reason, several radiographic views have been used to visualize the STJ in a clinical setting [2,3,23,32,33]. Nevertheless, no study has systematically evaluated which aspects of the three-dimensional anatomy of the STJ are visualized on two-dimensional radiographic views commonly used to view the hindfoot region. With the introduction of WB computed tomography (CT), a more detailed analysis of the STJ during loading became possible [8,9]. Although it is a promising technology for imaging the STJ, WB CT has not yet become a clinical standard. Moreover, to measure the STJ AoR, subjects must maintain the STJ neutral position. The STJ neutral position is the position typically used by clinicians to obtain the exact value of measurement. However, a CT scanner takes a lot of time to perform the measurement.

Furthermore, this imaging methodology is not widely available and involves time-intensive data postprocessing, access to specialized equipment is required, and it may be difficult to analyze easily. To supplement this insufficiency, the equipment is to be invented with “gold standards” since this study is based on direct measurements [4].

Thus, this study aimed (1) to determine the accuracy of the technique in vivo using a custom-made locator, which allows the tracking and measurement of the STJ inclination and deviation (Figure 1); (2) to test the validity and reliability of the custom-made locator; and (3) to determine whether the STJ AoR estimates consistent with previously reported (“gold standard”) ranges are found from the locator.

## 2. Materials and Methods

### 2.1. Participants

Thirty-eight healthy male participants without a history of lower extremity injuries were included in this study (age, 22.89 ± 9.11 years; weight, 77.68 ± 18.32 kg; height, 176.16 ± 14.16 cm). The exclusion criteria were as follows: individuals who had ankle surgery or nervous system damage or disorder and those with any injuries to the lower limbs within the past three months that could affect the neuromuscular function.

The purpose of this study and the experimental procedures were communicated verbally to the participants, and written informed consent was obtained from all participants before the experiment. This study was approved by the Institutional Review Board (IRB) of Yonsei University (IRB no. KISS-1806-034-01) in Seoul, South Korea, to comply with the ethical principles of the Declaration of Helsinki (1975, revised 1983).

### 2.2. Instruments

A custom-made STJ locator (STJL) was used in this study. For ease of measurement, a specially built STJL (Figure 2) consisting of an anterior exit pointer, posterior exit pointer, and basic aluminum bolt with a body manufactured from a light firm acrylic material was used. Lockable universal joints were used to allow movement in the desired direction, and pointers were used to identify the position of the STJ. Foot size measurement was set up from 190 mm to 330 mm, considering gender and ethnic differences. Similarly, the instrument angular measurable range was designed to reach up to ±60° to ease flexible measurement. The STJ AoR inclination is measured using a digital mini-protractor (WWC-TE, Inc., Seoul, Korea).

### 2.3. Testing Procedures (Validity and Reliability of the STJL for Measuring STJ Inclination)

We validated the device by measuring the STJ inclination angle recorded using a radiography machine (Median MDXP-40, Inc., Seoul, Korea), which was then compared to the value obtained from the custom-made STJL. All measurements were conducted after positioning and securing the subjects in an STJ-neutral position. The measurements were conducted over a day to minimize any errors. The STJ AoR inclinometer performed 7 years’ experience as a qualified athletic trainer and radiographic imaging taken 5 years’ experience as a radiological technologist.

During radiographic imaging, the subject’s feet were fixed in a tandem position (Figure 3), with a prescribed spacing between the front and rear leg stance for even weight distribution. In addition, all reference points were set to equal, considering the adjustment in measurement position among subjects and the error position of the measuring instrument. Radiographic imaging took a day for accurate measurement. All tasks were conducted at the same height on a custom-made table.

First, to locate the STJ AoR, an arbitrary point with the least movement in the anterior–posterior direction (Figure 4A,B) was identified, and lines were drawn. Then, the STJL was fitted on the subject under WB conditions, and two laser pointers were fixed at both exit points to identify the inclination angle of the STJ. The measurement of the STJ AoR was conducted on the lateral angle based on the horizontal and vertical planes. The anterior and posterior exit points were located near the talus antemedial and calcaneus post-lateral, respectively. Figure 4C,D show the position of the pointers at the anterior and posterior exits, respectively, and Figure 4E shows the lateral view. A digital inclinometer was then used to measure the angle of the STJ AoR in the transverse (Figure 4F) and sagittal (Figure 4G) planes. The normal STJ AoR has sagittal and transverse planes of 42° and 16°, respectively.

### 2.4. Data Analysis

In this study, the STJ AoR was measured using a custom-made STJL and radiography device (Median MDXP-40, Inc., Seoul, Korea). From the custom-made STJL, the angle was measured in the sagittal and transverse planes using a digital inclinometer after positioning and securing the pointers to the anterior and posterior exit points. An average was taken from the five trials. The radiographic measurements were conducted at “S” hospital and were used to measure the STJ AoR in the sagittal plane. Before the experiment, consultations with radiologists were conducted to avoid any possible errors while performing the measurements.

### 2.5. Statistical Analysis

The test–retest reliability of the data collected on the two days was analyzed for each plane, and the Pearson coefficient correlation (r) was used for calculating the concurrent validity using the Statistical Package for the Social Sciences, version 25 (IBM Corp., Armonk, NY, USA). The test–retest reliability of the custom-made STJL for measuring the sagittal and transverse STJ inclinations was performed using the intraclass correlation coefficient (ICC), standard error of measurement (SEM), and minimal detectable difference (MDD). Two ICC (2, 1) values, representing the agreement of five trials for each plane from the sagittal and transverse STJ inclinations, were computed. The ICC values were defined as “poor” when they were below 0.20, “fair” when they were from 0.21 to 0.40, “moderate” when they were from 0.41 to 0.60, “good” when they were from 0.61 to 0.80, and “very good” when they were from 0.81 to 1.0 [14]. The SEM was defined as standard deviation (SD) multiplied by the square root of the ICC subtracted from 1 [15]. Moreover, the MDD was analyzed to determine the minimum threshold of measurement to ensure that differences between measurements were real and outside the error range by multiplying the SEM by the square root of 2 [16]. To investigate the validation between the two devices, we used the Bland–Altman method (*p* < 0.05). The limits of agreement were set to ±1.96 SDs from the mean.

## 3. Results

### 3.1. Test–Retest Reliability

The test–retest reliability of the custom-made STJL in measuring the sagittal and transverse STJ inclinations with ICC, SEM, and MDD is described in Table 1. A high correlation with ICC (2, 1) values of 0.930 for the sagittal plane (very good) and 0.750 for the transverse plane (good) was observed.

The test–retest reliability results for the STJL were as follows: the SEM ranged from 0.56 to 0.78, and the MDD ranged from 1.57 to 2.16.

### 3.2. Construct Validity

The Bland–Altman plot (Figure 5) showed that most observations (>95%) were near the mean of the differences in the instruments for both phases (±1.96 SD range of the differences). This analysis indicates that both instruments present a high degree of agreement. The 95% limits of agreement ranged between −5.22 and 2.49 for the sagittal STJ inclination. The central line represents the absolute average difference between the instruments. Short-dashed lines represent the upper and lower 95% limits of agreement, respectively.

## 4. Discussion

For some years, clinicians have been studying methods to determine the clinical position of the STJ AoR [6]. Because surgical or conservative therapy to control irregular forces occurring symptomatologically could only be estimated [34,35,36], it is not known where the axis of the joint involved in the osseous structures is located. The transition of the STJ AoR could result in diverse degrees of abnormal STJ pronation or supination accompanied by various symptomatologies. This study proposes to confirm the clinical employment of the STJL and conduct follow-up studies related to the tendencies of the STJ AoR. Radiographic imaging is the recommended method for measuring the sagittal inclination of the STJ AoR [4]. The main conclusion of this study is that the custom-made STJL has reliability and validity values comparable to those of radiographic imaging (Table 2). The results were consistent with this hypothesis.

In the past, various studies have been reported in order to pursue the inclination of STJ AoR [5,10,13,16,17,18,19,21,24,25,26,27,28,37,38,39,40]. Nevertheless, it is considerably difficult to measure the SJT AoR precisely and conveniently in a clinical setting. In this study, a novel instrument was used to measure the inclination of the STJ AoR through the information suggested by this author. Furthermore, the instrument was made of aluminum to reduce the margin of error from minute movements in WB conditions. The degree of error connecting the axis to the sagittal and transverse planes was reduced during the measurement of the STJ inclinations. In particular, a digital inclinometer (Figure 2) was used to measure up to 0.1° units for elaborate inclination. As a method for measuring the inclination on each plane of the STJ AoR, it was measured using the method presented in (Figure 4).

Moreover, the inclination of the transverse plane was estimated by locating the direction of the STJL into the direction of the frontal plane after primarily measuring the inclination of the sagittal plane. Despite the absence of a reference to compare the inclination of the transverse plane to, the inclination of the sagittal plane supports the result of this study when compared with the measurements obtained using a previous method [4].

The results show high intra-rater reliability using the radiographic images with four measured angles during WB positions, in accordance to the “gold standard.” Among the four kinds of measurement methods introduced by McClay, the second method presented the lowest SD. This occurs when there is less error in discerning the related anatomic index partially. Morris and Jones later reported that the position of the STJ AoR is formed from the anteromedial aspects of the talus to the posterolateral aspects of the calcaneus. In addition, the surface of the skin where the least movements are generated could be identified from the axis location while the feet move. In this study, the position of the STJ AoR was analyzed using McClay’s radiation imaging technique based on the study by Morris and Jones. Regarding the reliability analysis, this study resulted in highly reliable measurements during re-examination, since the ICC values were “very good” (sagittal ICC: 0.93). This range was similar to the results reported by McClay (sagittal ICC: 0.98), implying high accuracy and small error similar to that of previous studies.

The validity of radiographic imaging and the STJL were investigated using the angle.2 methods developed by McClay. This range was close to that reported by Manter (42 ± x (not reported)), Root (41 ± 8.36), and Isman and Inman (41 ± 9). The range of the sagittal STJ inclination was similar to that reported in a previous study [4]. While it was measured in a non-WB condition using a cadaver in a previous study, the STJ was measured in a WB condition (neutral STJ position) in this study. This becomes an essential baseline for determining measurement consistency. In addition, consistent evaluation is significant, as identifying the accurate location of the axis is the standard in evaluating foot models [22,41]. In addition, identifying the specific estimates of STJ AoR could provide an accurate estimation of joint moments, joint angles, and muscle forces [5], as well as improving the clinical assessment of the ankle.

This study has some limitations. First, radiation imaging technology is limited. Recently, for proper assessment of the STJ, a Harries–Beth view, broadened view, or lateral oblique axial projection can be employed (Lopez–Ben imaging of the STJ). However, most methods are used to measure the tension of the ligament by the instability of the STJ. Second, the analysis of STJ AoR images is conducted using CT or magnetic resonance imaging rather than plain radiography [24,25,26,27,28].

However, these methods could only be used in non-WB conditions. In most aforementioned studies, information on the standard and reference of the STJ AoR was inadequate. Furthermore, most studies analyzing the STJ AoR have introduced imaging technology used in surgery, and no discussion exists regarding the methods that can be conveniently used in clinical settings.

In this study, technical limitations, such as indirect transverse plane measurements using the STJL, were overcome. These results are highly similar to those of descriptive studies that were used as the starting point of analysis. However, measuring the transverse plane of the STJ is uncommon in clinical practice since the evidence on the reliability and validity of the transverse plane is insufficient.

## 5. Conclusions

The custom-made STJL may be used in the clinical setting since its validity and ICC were high, indicating consistent measurements. Along with the measurements using the custom-made STJL, further studies including motion analysis are warranted to provide more information regarding the relationship between STJ AoR inclinations and STJ movements.

## Figures and Tables

**Figure 1 ijerph-18-05494-f001:**
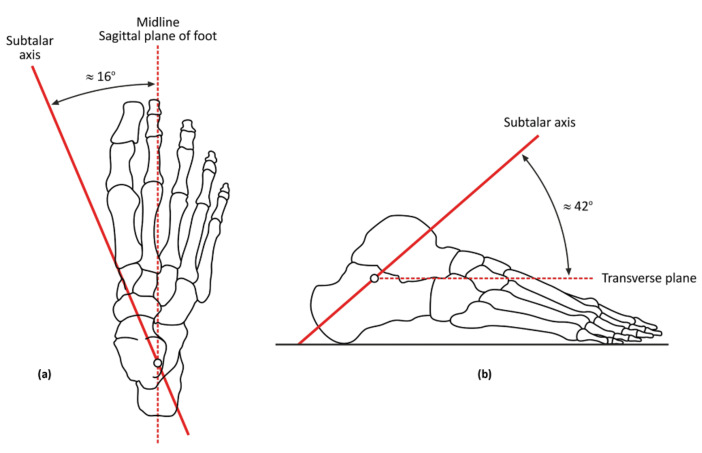
(**a**) Inclination of STJ in transverse plane and (**b**) inclination of STJ in sagittal plane.

**Figure 2 ijerph-18-05494-f002:**
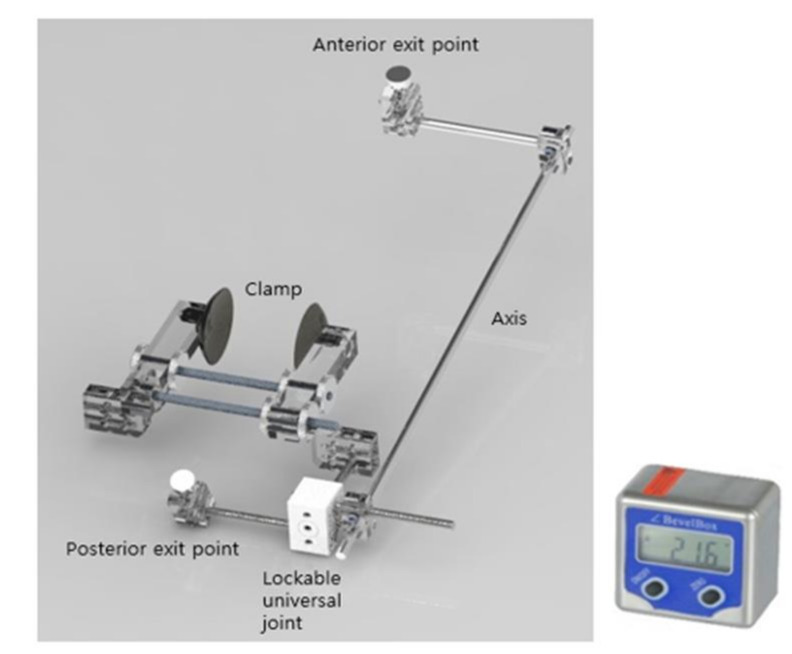
Structure of the custom-made STJL (**left**). Digital mini-protractor (**right**).

**Figure 3 ijerph-18-05494-f003:**
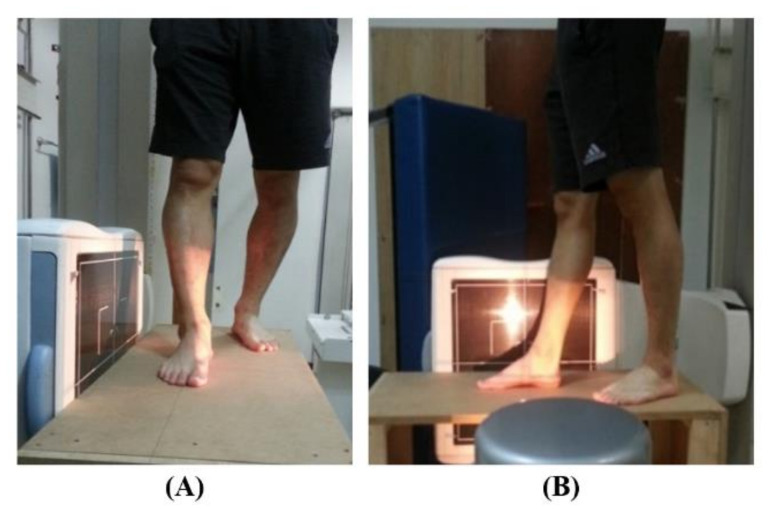
(**A**) Anterior and (**B**) lateral views of the subject while measuring the subtalar joint axis of rotation inclination in the sagittal plane using radiography.

**Figure 4 ijerph-18-05494-f004:**
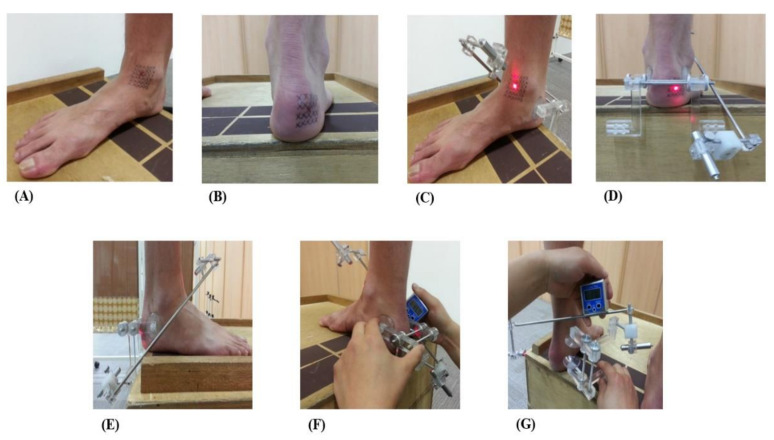
The process of locating the subtalar joint (STJ) axis of rotation inclination angle using the custom-made STJ locator (**A**) Anterior view, (**B**) Posterior view, (**C**) Anterior pointer, (**D**) Posterior pointer, (**E**) Lateral view, (**F**) STJ inclination, (**G**) STJ deviation.

**Figure 5 ijerph-18-05494-f005:**
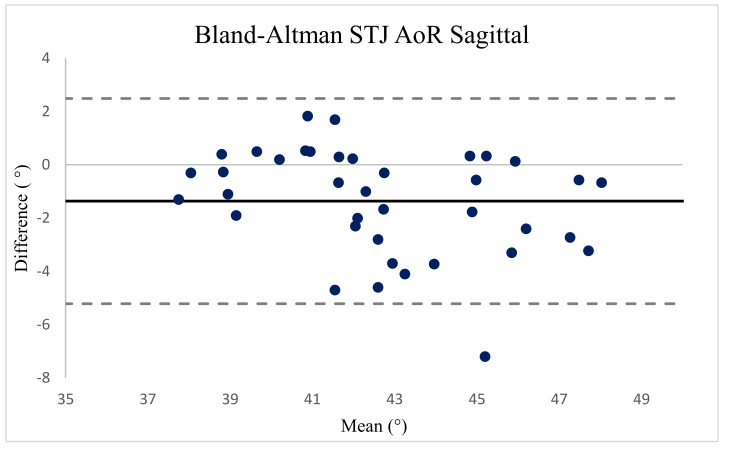
The Bland–Altman plot presenting the results with a significantly high correlation coefficient. AoR, axis of rotation; STJ, subtalar joint.

**Table 1 ijerph-18-05494-t001:** Means ± standard deviations and reliability (intraclass correlation coefficient (2, 1) and standard error of measurement (SEM)) and minimal detectable difference (MDD) values for each subtalar joint inclination (sagittal and transverse) collected during Days 1 and 2. The descriptive statistics, SEM, and MDD are reported in degrees.

Static Measurement	Day 1	Day 2	ICC	SEM	MDD
Sagittal STJ inclination	42.22 ± 1.79	42.86 ± 1.91	0.93	0.78	2.16
Transverse STJ inclination	16.15 ± 1.65	16.10 ± 1.12	0.75	0.56	1.57

ICC, intraclass correlation coefficient; MDD, minimal detectable difference; SEM, standard error of measurement; STJ, subtalar joint.

**Table 2 ijerph-18-05494-t002:** Pearson’s correlation coefficient between the radiographic image and subtalar joint locator (*p* < 0.01).

	Pearson Correlation (0.798)	
	Radiographic image	STJL
M ± SD	42.50 ± 2.76	43.58 ± 3.23

STJL, subtalar joint locator.

## Data Availability

Data are available on request.

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
