# Peer review of "Validity and Reliability of a Novel Instrument for the Measurement of Subtalar Joint Axis of Rotation"

_ijerph, 2021, doi:10.3390/ijerph18105494_

Round 1

Reviewer 1 Report

The study proposed a novel instrument for the measurement of subtalar joint axis of rotation. The Authors described the structure of the device and performed a test of its validity/reliability with 38 participants. The results from the device were compared to the ones obtained with radiography - a currently recommended method for measuring the axis. Furthermore, the results were visualized and analyzed using common statistical approaches, such as Bland–Altman plot.

The topic itself is not entirely new, but the device and the results obtained by the Authors may have clinical applications. Furthermore, the Authors argue that while some previous studies were performed, the problem is still very much open.

1. I would advise to add additional figures, mainly to the introduction - regarding the joint and the angles. The subject of the study is complex and without proper visualizations, the main ideas might be hard to understand for most readers of the journal.

2. Fig. 1 must be of higher quality. Larger font (comparable to the one in the text) and higher resolution.

Author Response

Dear Reviewer

First of all, we thank you for your kind comment.

We also appreciate the time and effort you have dedicated to providing insightful feedback on ways to strengthen our paper. Thus, it is with great pleasure that we resubmit our article for further consideration. W have incorporated changes that reflect the detailed suggestions you have graciously provided. We also hope that our edits and the responses we provide below satisfactorily address all the issues and concerns you have noted.

To facilitate your review of our revisions, the following is a point by point response to the questions and comments delivered in your latter dated.

  1. I would advise to add additional figures, mainly to the introduction - regarding the joint and the angles. The subject of the study is complex and without proper visualizations, the main ideas might be hard to understand for most readers of the journal

Answer: We have added a figure regarding inclination of subtalar joint axis

  1. Fig. 1 must be of higher quality. Larger font (comparable to the one in the text) and higher resolution.

Answer: We have made a new figure (Subtalar joint locator) with font using by illustrator.

Additionally, we have corrected each figure number.

We look forward to hearing from you at your earliest convenience.

Best regards,

Reviewer 2 Report

The manuscript “Validity and Reliability of a Novel Instrument for the Measurement of Subtalar Joint Axis of Rotation” by Byonghun Kim  and Saeyong Lee is an article that aimed o determine the accuracy of the technique in vivo using a custom-made locator, which allows tracking and measurement of the STJ inclination and deviation and to test the validity and reliability.

Introduction

  1. Literature review of clinical methods [10-20] insufficient. The latest work from 2003 and the rest even from 1941? And this methods -https://doi.org/10.1038/s41598-020-57912-z

Methods

  1. Patients - healthy male participants without a history of lower extremity injuries were included. Are the described inclusion criteria correct or are they exclusion criteria?
  2. It was not described who made the measurements and what his experience was.
  3. When were the tests performed? Author's research published as a 2014 conference report

Results

  1. There is no comparison to CT

Discussion

  1. “number of equipment used to measure the angle on the feet and the motion of the STJ has increased” Please give examples.
  2. References older than 5 years – only 1 reference from 2017
  3. Is the proposed method is so simple and time-consuming to apply it every day?

Author Response

Dear Reviewer

First of all, we thank you for your kind comment.

We also appreciate the time and effort you have dedicated to providing insightful feedback on ways to strengthen our paper. Thus, it is with great pleasure that we resubmit our article for further consideration. W have incorporated changes that reflect the detailed suggestions you have graciously provided. We also hope that our edits and the responses we provide below satisfactorily address all the issues and concerns you have noted.

To facilitate your review of our revisions, the following is a point by point response to the questions and comments delivered in your latter dated.

Introduction

  1. Literature review of clinical methods [10-20] insufficient. The latest work from 2003 and the rest even from 1941? And this methods -https://doi.org/10.1038/s41598-020-57912-z

Answer: We have redrafted the Introduction lines (52 to 55) to establish a clearer focus.

Methods

  1. Patients - healthy male participants without a history of lower extremity injuries were included. Are the described inclusion criteria correct or are they exclusion criteria?

Answer: We have revised the text “inclusion” to “exclusion” line 83.

  1. It was not described who made the measurements and what his experience was.

Answer: We have added a sentence line 107 to 109

  1. When were the tests performed? Author's research published as a 2014 conference report

Answer: The tests performed at 2014 and I have submitted the Journal of foot and ankle research as abstract. However, I have not published the study.

Results

  1. There is no comparison to CT

Answer: We agree with comparing to CT however, most of CT scanner can not measure in a weight-bearing condition. Recent studies have shown the CT scanner (PedCAT, Curvebeam, Warrington, USA) that can measure in a weight-bearing condition however, measuring CT scanner is needed more time than radiography image. Also, all measurements of STJ AoR needed the subjects in a STJ neutral position.

Discussion

  1. “number of equipment used to measure the angle on the feet and the motion of the STJ has increased” Please give examples.

Answer: We have rewritten the text line from (200 to 202) to (202 to 204).

  1. References older than 5 years – only 1 reference from 2017

Answer: We have added 5 more reference from 2017. Lines (188, 238 and 239).

  1. Is the proposed method is so simple and time-consuming to apply it every day?

Answer: Yes, it is very convenient method if assessors who were assumed the reliability. It takes time less than few minutes to measure.

We look forward to hearing from you at your earliest convenience.

Best regards,

Round 2

Reviewer 2 Report

Thank you for the explanations and supplements.

1. The lack of comparative testing to the CT scanner with weight-bearing measuring is a limitation. The test itself is obviously more time consuming, but the test should be used to confirm the equivalence of this simple measurement method.

2. Does the radiological technologist have the same clinical experience as, for example, a radiological technician?

Author Response

Dear Reviewer

First of all, we thank you for your kind comment again.

We also appreciate the time and effort you have dedicated to providing insightful feedback on ways to strengthen our paper. Thus, it is with great pleasure that we resubmit our article for further consideration. We have incorporated changes that reflect the detailed suggestions you have graciously provided. We also hope that our edits and the responses we provide below satisfactorily address all the issues and concerns you have noted.

To facilitate your review of our revisions, the following is a point by point response to the questions and comments delivered in your latter dated.

  1. The lack of comparative testing to the CT scanner with weight-bearing measuring is a limitation. The test itself is obviously more time consuming, but the test should be used to confirm the equivalence of this simple measurement method.

         Answer: We have redrafted the limitation of testing to the CT scanner line                         (68 to 71).

  1. Does the radiological technologist have the same clinical experience as, for example, a radiological technician?

          Answer: Yes, correct. It has been mentioned line (115 to 117) and (148 to                           150).

We look forward to hearing from you at your earliest convenience.

Best regards,
